

# CNN for image-based sediment detection applied to a large terrestrial and airborne dataset

Xingyu Chen[1,2], Marwan A. Hassan[2] and Xudong Fu[1]

[1]Department of Hydraulic Engineering, State Key Laboratory of Hydroscience and Engineering, Tsinghua University, Beijing, China
[2]Department of Geography, The University of British Columbia, Vancouver, BC, Canada.

*Correspondence to*: Xudong Fu (xdfu@tsinghua.edu.cn)

**Abstract**: Image-based grain sizing has been used to measure grain size more efficiently compared to traditional methods (e.g. sieving and Wolman pebble count). However, current methods (e.g. BASEGRAIN) are largely based on detecting grain interstices from image intensity which not only require a significant level of expertise for parameter tuning but also underperform when they are applied to sub-optimal environments (e.g. dense organic debris, various sediment lithology). We proposed a model (GrainID) based on convolutional neural networks to measure grain size in a diverse range of fluvial environments. A data set of more than 125,000 grains from flume and field measurements were compiled to develop GrainID. Tests were performed to compare the predictive ability of GrainID with sieving, manual labeling, Wolman pebble counts and BASEGRAIN. When compared with the sieving results for a sandy-gravel bed, GrainID yielded high predictive accuracy (comparable to the performance of manual labeling) and outperformed BASEGRAIN and Wolman Pebble counts (especially for small grains). For the entire evaluation dataset, GrainID once again showed fewer predictive errors and significantly lower variation in results in comparison to BASEGRAIN and Wolman pebble counts and maintained this advantage even in uncalibrated rivers with drone images. Moreover, the existence of vegetation and noise have little influence on the performance of GrainID. Analysis indicated that GrainID performed optimally when the image resolution is higher than 1.8 mm/pixel, the image tile size is 512*512 pixel*pixel and the grain area truncation values (the area of smallest detectable grains) were equal to 18 - 25 pixels.

## 1 Introduction

Sediment grain size and its spatial variability are fundamental in river dynamics (e.g. sediment transport, channel evolution), ecological studies (e.g. aquatic habitat; fishery) and river restoration engineering. However, the measurement of grain size has been time-consuming and laborious especially in mountain rivers due to the wide range of grain size classes, diverse grain lithology, the hiding of grains, diverse structures and the influence of organic materials. The most widely used grain-sizing method is sieving (Kellerhals and Bray, 1971) and is used as a benchmark to other methods when reliable sediment samples are able to be collected (Church et al., 1987). Wolman (1954) proposed a pebble count method (Wolman method) that samples a minimum of 100 pebbles from the riverbed surface with a grid-based system. Limited to material > 8 mm (Kellerhals and





Bray, 1971), the Wolman method has been especially popular in the field due to the limited equipment required and its benefit of reducing sampling times whilst providing a relatively valid estimation of reach-scale grain-size distribution. Since then, various versions of the Wolman method were proposed with different approaches to collecting stones such as the random walk approach for particle collection (Leopold, 1970), superimposing gravel templates upon the sedimentological unit for reduced

operator error (Bunte and Abt, 2001), and image-based Wolman method analysis (Hassan et al., 2020; An et al., 2021).

Since the 1970s, advances in high resolution photography have provided scientists the opportunity to estimate sediment grain size in river beds from images, largely reducing sampling time for large-scale field surveys compared to sieving and Wolman methods (Church et al., 1987; Adams, 1979). However, the development of such methods to measure grain size from images has been challenging as early studies relied on the laborious manual identification of grain boundaries on vertical images

(Adams, 1979; Ibbeken and Schleyer, 1986) and only within the last 20 years has there been the development of automated grain sizing algorithms (Graham et al., 2005b; Buscombe et al., 2010). Generally, image-based automated grain sizing methods can be classified from percentile-based to object-based methods (Buscombe, 2020). Percentile-based methods (Carbonneau et al., 2004; Rubin, 2004; Buscombe, 2020; Buscombe et al., 2010) estimate grain size distribution based on statistical analysis of image intensity and texture through pixel-wise simple autocorrelation algorithms (Rubin, 2004), grain size prediction as a

function of both local image texture and semi variance (Carbonneau et al., 2004), spectral decomposition of an image (Buscombe et al., 2010) and convolutional neural networks (CNN) (Buscombe, 2020; Mueller, 2019; Lang et al., 2021). Object-based methods (Sime and Ferguson, 2003; Detert and Weitbrecht, 2012; Graham et al., 2005a; Graham et al., 2005b; Mcewan et al., 2000) apply sequences of grain separation algorithms to detect grain interstices and identify each individual grain on the bed. Mcewan et al. (2000) applied an automatic edge-detection algorithm on Digital Elevation Models (DEMs)

of grain surfaces generated by laser scanning and reported promising grain-size measuring results. Sime and Ferguson (2003) presented a modified edge-detection algorithm which combined both edges seeding and partial watershed segmentation algorithms. *Graham et al.* (2005a, 2005b) proposed a double threshold interstice-detection approach in which the threshold levels to detect grain interstices were initially defined based on image intensity distribution and further refined through a bottom-hat filter. Based upon this approach, Detert and Weitbrecht (2012) proposed an enhanced grain detecting model (named

as BASEGRAIN) which applies a five-step image-processing procedure to separate grains on the bed.

As noted by several researchers (e.g., Carbonneau et al. (2004), Graham et al. (2010) and Buscombe (2020)), object-based methods require sophisticated object segmentation algorithms and theoretically cannot be used on grains smaller than one pixel, however, object-based methods can provide grain-scale information on spatial variability which is essential in not only predicting but also understanding the processes of flow resistance (Chen et al., 2020), sediment transport (Yager et al., 2018)

and aquatic habitat (Reid et al., 2020). The BASEGRAIN model developed by ETH Zurich is a state-of-art object-based grain sizing software, but it requires extensive parameter tuning (the model contains more than 40 adjustable parameters and seven key parameters) and a significant level of expertise to be applied to sub-optimally captured images. Moreover, the model only focuses on detecting edges and as such performs poorly in fluvial environments where dense organic debris, various sediment lithology, and non-uniform lighting are present in the photo (Detert and Weitbrecht, 2020). The limitations of BASEGRAIN





in these suboptimal environmental conditions can be overcome using Convolutional Neural Networks (CNN) which have been extensively used in computer vision (Krizhevsky et al., 2012) and biomedical applications (Ronneberger et al., 2015). Through repeated convolutions and pooling on the input images, CNN can automatically capture not only object edges but also high-level features such as shape, color and texture (Buscombe, 2020). In addition, with nonlinear activation functions (e.g. sigmoid) in every neuron, the network is capable of learning the nonlinearity of grain features under diverse environments. When trained

with large sets of images, CNN has proven to be a robust tool for object classification and identification even when applied to sub-optimally conditioned images (He et al., 2016).

For image segmentation tasks, the state-of-art CNN architecture is *U-Net* (Ronneberger et al., 2015), which was designed to separate individual cells in biomedical images. *U-Net* has been successfully applied to solve many problems such as multi-organ segmentation (Oktay et al. (2018), detection of lung abnormalities (Kohl et al. (2018) and autonomous driving (Tran

and Le (2019). The detection of grains is different with the tasks above in regards to the wide range of grain size classes, diverse grain lithology and the hiding of the grains. The size of large sediments to be detected can be several magnitude larger than the size of fine sediments. Mueller (2019) tried applying *U-Net* to detect grains, however, the paper only presented one prediction result for a gravel-bed picture and no systematic analysis was reported. The potential of *U-Net* to detect sediments in diverse fluvial environments has not yet been studied. Meanwhile, when *U-Net* applications are applied to large-scale

orthophotographs (e.g. images acquired by unmanned aerial vehicles (UAVs) and structure from motion (SFM) photogrammetry), the scale and resolution of input images were limited by GPU memory and model complexity. As such, predictive errors are introduced when splitting the large images into sub-tiles for easier processing. As a result, how can we reduce errors when applying *U-Net* for grain detection in a diverse range of fluvial environments? How does image resolution and image tile size influence the predictive ability of *U-Net*? What is the size of the smallest detectable grain unit for *U-Net*?

These questions have yet to been answered. Therefore, it is of great value to develop a *U-net*-based model for grain size measurement in diverse fluvial environments.

In this paper, we propose a model (GrainID) based on *U-Net* with an overlap-tile strategy to detect grain size from images in a diverse range of fluvial environments. To achieve our goal, we (i) compiled a large dataset of grain images containing more than 125,000 grains in a diverse range of fluvial environments and trained GrainID with the datasets, (ii) compared the results

of GrainID with sieving, manual labeling, Wolman and BASEGRAIN methods, (iii) tested the performance of GrainID for uncalibrated rivers with airborne photos, and (iv) evaluated the influence of vegetation, inter-granular noise, image tile size, and resolution on model performance.

## 2 Data

The datasets (84 flume, 118 field photos) cover a wide range of fluvial environments and include a variety of field site and

flume experiment images. As shown in Table 1, the datasets were grouped into three subsets according to channel conditions: (1) Flume channel; (2) Forested mountain rivers; and (3) Sparsely vegetated large rivers.



*Flume channel*: The first flume set (SAFL dataset) consists of 51 sediment photos collected in a riffle-pool experiment (Fig. 1a; flume size: 2.8m * 55m) carried out in the St. Anthony Falls Laboratory (SAFL) at the University of Minnesota (Singh et al., 2013). The channel bed samples were primarily composed of a sandy-gravel mixture created by adding sand to the clean gravel mixture and turning the bed over. Bed surface samples were then collected and sieved using the Klingeman Sampling protocol (Kondolf, 2000; Klingeman and Emmett, 1982). The second flume set (MCHEL dataset) consists of 33 flume photos taken from a step-pool experiment (flume size: 0.4m*5m) carried out in the Mountain Channel Hydraulic Experimental Laboratory (MCHEL) at The University of British Columbia (Fig. 1b). A non-uniform sediment mixture with a wide grain size distribution between 0.5 and 64 mm (measured by sieving) was used. The sediments in MCHEL are painted in different colors to classify different grain-sizes, but the issue of wearing on the grain surface introduces inter-granular noise like the noise introduced by different grain lithologies in the field.

*Forested mountain rivers*: 70 grain photos (Brayshaw, 2012; Helm et al., 2020) were collected in 18 small forested gravel-bed rivers (basin area < 100 km$^2$; Fig. 1c) in British Columbia, Canada. Visual assessments suggest that a large proportion of the channels were hidden beneath a dense forest canopy composed of both coniferous and deciduous tree species (Fig. 1c), with a channel slope ranging from 0.007 to 0.184, and the sediments cover a wide range of sedimentary, metamorphic, intrusive and extrusive lithologies (Brayshaw, 2012; Hassan et al., 2014). The grain size information in Table 1 for forested rivers was calculated by Brayshaw (2012) using the Digital Gravelometer software proposed in Graham et al. (2005b).

*Sparsely vegetated large rivers*: Six UAV photos were collected by our research group in two large mountain rivers: Upper Yangtze River (Fig. 1d) and Yaluzangbu River from China. The photos were taken along the riverbank in which they were influenced by the presence of water, waves and cohesive sediments. There was sparse vegetation in the images and the sediments were primarily composed of moderately weathered silicate mineral. We also specifically collected 42 photos primarily consisting of environmental elements (e.g. wood, vegetation and water) with limited sediment grains in the images to train our machine learning model to better distinguish sediments from these environmental elements.

The datasets of 202 images were randomly split into two subsets (Table 1): a training subset with 136 images and a test subset with 66 images. To evaluate the influence of vegetation and inter-granular noise on model performance, the test subset was further grouped based on the presence of vegetation and inter-granular noise, in which GrainID, BASEGRAIN and the Wolman method were tested for each of the data groups. As shown in Table 1, the tested images with / without vegetation were marked with the superscript [v] / [nv] while the tested images with/without inter-granular noise were marked with the superscript [i] / [ni].

## 3 Methods

### 3.1 Manual labeling

Manual labels were created for all grain images as baseline labels to train and evaluate the methods. Manual labeling was chosen as it is a robust method when applied to diverse fluvial environments due to its basis on human cognitive analysis and it has been widely used as baseline method for grain detection studies (Sime and Ferguson, 2003; Graham et al., 2005a;



Ronneberger et al., 2015). Figure 2a-2d are the examples of two field images and the corresponding manual labels. Fig. 2a

shows a bed with vegetation (Anderson Creek), and Fig. 2c shows a bed without vegetation but with inter-granular noise due to grain lithology. The grains were marked as white pixels isolated from each other and the interstices are marked as black pixels (Fig. 2b, 2d). For grains covered by vegetation, only the exposed part was labeled, and grains with area of 23 pixels were chosen as the smallest grains to be labeled (Detert and Weitbrecht, 2012). As shown in Fig. 2, the images are large enough to capture the grain size distribution even with the presence of vegetation in the image. A total of 128461 grains were

marked for the entire dataset of 203 images (67612 in the training datasets, 60849 in the test datasets) by two masters, in which operator-1 created 170 images and operator-2 created 33 images. When an operator finished labeling an image, the labels would be checked by the other operator, so that all the labels used in this paper were confirmed by two operators.

To explore the consistency in labeling and estimate human related errors, five human masters (including operator 1 and 2) were asked to label a fixed dataset of 12 photos representing diverse fluvial environments. The photos are selected from Table

1, in which six are from *Forested mountain rivers*, three are from the SAFL dataset (Singh et al., 2013), two are from the MCHEL dataset, and one is an airborne photo from Yaluzangbu River.

Boxplots were applied to describe the variation of predicted grain size between the operators. The boxplot displays the five-number summary of a set of data including the maximum, third quartile, median, first quartile, and minimum (from top to bottom). Figure 3 shows the boxplot of normalized grain size $D_{normalized}$ ($D_{normalized} = (D - D_{mean}) / D_{std}$) for different grain

percentiles and different operators, in which $D$ is the predicted grain size for a manual label, $D_{mean}$ is the mean grain size value of 12 photos chosen for analysis, $D_{std}$ is the standard deviation of grain size value of the 12 photos. As shown in Fig. 3, the five operators showed a consistent prediction for all $D_{normalized}$ statistics and all grain percentiles. An exception is $D_{50}$, in which operators 2 and 5 showed a higher maximum value of $D_{normalized}$ than the other three operators. The inconsistency for $D_{50}$ prediction mainly arises from the predictions for the three photos from the SAFL dataset in which the bed contains a lot of fine

grains, and operators 2 and 5 overestimated the $D_{50}$ by merging fine grains as larger sediments. As such, the analysis suggests the manual labels datasets prepared by operators 1 and 2 were consistent with labels created by human masters.

## 3.2 GrainID

### 3.2.1 Model framework

A model framework (GrainID) to detect grains from images in diverse fluvial environment was introduced in this section. Fig.

4a shows the framework of the GrainID model working in its 3-step procedure, and Table 2 lists the detailed description of each processing step. For image pre-processing (*step* 1), we tried three image filters in the Python Image Processing Library: *pillow* (python package): *edge enhancement*, *sigmoid contrast,* and *detail*, in which the *Sigmoid contrast* filter was chosen for its better overall performance. Image augmentation (Fig. 4c), a widely-used technique for CNN prediction, allowed the network to learn variances in object location, rotation or deformation without the need to see these transformations in the annotated

image corpus.

The CNN prediction for the border region of an image is invalid as the convolution used mirroring context rather than real image information of the border for prediction purposes (Ronneberger et al., 2015). As such, errors are introduced when splitting a large photo into many image tiles for U-net prediction. To solve this problem and achieve seamless prediction, in step '*Image extrapolation-2*' and '*Image split*', we applied an overlap-tile strategy (Ronneberger et al., 2015). The overlap-tile

strategy only utilizes the central parts of an image tile to be used for valid prediction. For example (Fig. 4b), for image tiles (red and blue dash rectangles, 512*512 pixels) used for U-net inputs, only the center parts of U-net outputs (red and blue solid rectangles, 256*256 pixels) were accepted for predictions. To achieve seamless prediction, we created overlapping image tiles for our *U-net* inputs as dashed red and blue rectangles in Fig. 4b in step '*image split*', and the missing context in the border region was extrapolated by mirroring the border region in step '*image extrapolation-2*' (shown as shadow region in the image

in Fig. 4b).

In *step* 2, image tiles created by our overlap strategy were then input into *U-Net* for prediction. The predictions were then recombined into a full image. The final CNN prediction was calculated as a result assembly voted by predictions of the five augmented images, in which the voting rule was that a pixel will be calculated as an interstice if two or more predictions identify an interstice at that pixel so that the model can detect grain interstice and separate grains as much as possible. Four

post-processing algorithms were performed in step 3 in which holes inside grains were filled and grains with area < 20 pixels were filtered. To compensate the error of wide interstices due to human labeling, the interstices between grains were narrowed for 2 pixels using an inverse watershed algorithm. Finally, to further separate the merged grains, a watershed algorithm was performed based on grain centroid information.

### 3.2.2 U-Net: a CNN architecture for image segmentation

*U-Net*, evolved from CNNs, is specifically designed for image segmentation application. As shown in Fig. 4d, the U-shaped model architecture consists of two major paths: the contracting path (left part) and the expansive path (right part). The contracting path, similar to the typical CNN architecture consists of a sequence of 3*3 convolution layers for feature extraction and 2*2 max pooling layers for down-sampling. In the expansive path, every operation consists of a transposed convolution layer for up-sampling and two subsequent 3*3 convolution layers, where the transposed convolution layer expands the image

and maintains the same connectivity as the regular convolution. With this architecture, *U-Net* can maintain a consistent image size between the output and input and detect specific objects by doing classification on every pixel.

### 3.3 Manual sieving, BASEGRAIN and Wolman methods

The model proposed in this paper was compared to the manual sieving, Wolman and BASEGRAIN methods. The three methods were considered because they are widely used and accessible.

The manual sieving method was applied to bed samples from the SAFL dataset. Sediment samples were first weighed for a total mass and then sieved through a sieve set (mm): 32; 22.6; 16; 11.3; 8; 5.6; 4; 2.83; 2; 1.4; 1. The sediments of each sieve



as well as the fine sediments left in the pan were weighed once again, and the weight percentage of each size fraction were calculated (Singh et al., 2013).

The image-based Wolman method samples 100 grains based on an equidistant grid on the image where the sediment distribution was calculated via a grid-by-number approach that has been applied in many literatures (Kellerhals and Bray, 1971; Hassan et al., 2020).

The BASEGRAIN applies a five-step image processing algorithm to detect grains (Detert and Weitbrecht, 2012): In step (1) – (3), the model sequentially applies the (1) double grayscale threshold, (2) morphological bottom-hat transformations and (3) the Canny and the Sobel methods to detect grain interstices. In step (4), an improved watershed algorithm is performed for grain segmentation. In step (5), grains with an area < ~23 pixels are excluded and grain properties (e.g. a-axis, b-axis, orientation) are calculated. We followed the user guide (Detert and Weitbrecht, 2013) when using BASEGRAIN for our analysis in which the model parameter tuning process was done mainly by adjusting seven key parameters (parameters *medfiltsiz10*, *facgrayhr1*, *facgrayhr2*, *medfiltsiz20*, *criteriCutL2*, *areaCutLfA* and *areaCutWW* as introduced in (Detert and Weitbrecht, 2013)) to get the best visual segmentation of the grains.

## 3.4 Model evaluation

The grain size distribution was calculated for the images in the test datasets (Table 1). The a-axis (major-axis) of a grain was defined as the maximum Euclidean distance between two points on the grain boundary, and the b-axis (minor-axis) was calculated as the maximum intercept to the grain along a line perpendicular to the a-axis. Based on the b-axis and grid-by-area method (Kellerhals and Bray, 1971), sediment percentiles $D_5$, $D_{16}$, $D_{50}$, $D_{84}$ and $D_{95}$ were calculated for the results of manual labeling, GrainID and BASEGRAIN. The sediment percentiles of the Wolman method were calculated based on a grid-by-number method equivalent to the grid-by-area method demonstrated by Kellerhals and Bray (1971).

Predictive error for grain percentile $D_i$ for a tested image is defined as,

$$Err_i = \mathrm{abs}(1 - (D_{i,\ predicted} / D_{i,\ baseline})) \tag{1}$$

where $D_{i,baseline}$ and $D_{i,predicted}$ denote the baseline value and predicted value of $D_i$, abs() denote the absolute value.

Mean and median predicting error are used to evaluate the performance of different methods, where $Err_{i,\ mean}$ and $Err_{i,\ median}$ are mean value and median value of $Err_i$ for photos in the test datasets. Variation of predictions were measured in two ways, of which $V_{i,\ 3rd-1st}$ and $V_{i,\ max-min}$ denote the variations of *third quartile – first quartile* and *maximum - minimum* for $D_i$.

## 4 Evaluating the predicting ability of image-based grain sizing methods in diverse fluvial environments

In this section, we first compared the predictive ability of four image-based methods to manual sieving as it has been established as the most reliable grain sizing method (section 4.1). Subsequently, we tested the predictive abilities of GrainID in diverse environments based on the entire test dataset (section 4.2). Then, we tested the applicability and robustness of GrainID to uncalibrated rivers with airborne photos (section 4.3). Finally, we evaluated the influence of vegetation and inter-



granular noise on model performance (section 4.4). In, section 4.1, the manual sieving method was used as our baseline measurement. For the analysis in section 4.2, 4.3 and 4.4, manual sieving data was unavailable for the field datasets. The manual labeling was also used as baseline method, as the method is a robust grain sizing method and has also been widely used as a baseline method for grain detection studies (Sime and Ferguson, 2003; Graham et al., 2005a; Ronneberger et al., 2015).

### 4.1 Performance compared to sieving method

The dataset from SAFL (Singh et al., 2013) was compiled to evaluate the performance of image-based methods compared to the manual sieving method. Figure 5a–5d show a sample photo of the flume bed (Fig. 5a), the labels of Manual labeling (Fig. 5b), GrainID (Fig. 5c) and BASEGRAIN (Fig. 5d) predictions. As shown in Fig. 5a, the flume bed contains a lot of fine sediments. GrainID can predict sediment of a wide range of different sizes, whilst BASEGRAIN performs well for large grains but fails to predict fine grains.

The statistical analysis shows that, for small grains ($D_5$, $D_{16}$, $D_{50}$), manual labeling ($Err_{i, median}$ = 0.17, 0.10, 0.15) and GrainID ($Err_{i, median}$ = 0.16, 0.16, 0.16) significantly outperform BASEGRAIN ($Err_{i, median}$ = 0.72, 0.50, 0.30) and Wolman classification methods ($Err_{i, median}$ = 0.43, 0.46, 0.26). BASEGRAIN shows much larger variation than the other three methods (Fig. 6a) in terms of $V_{i, 3rd-1st}$. For large grains ($D_{84}$, $D_{95}$), the four methods show similar performance in terms of both $Err_{i, median}$ and $V_{i, 3rd-1st}$. BASEGRAIN consistently overestimated whilst the Wolman method consistently underestimated grain size for all percentiles. Overall, BASEGRAIN shows the worst performance, whilst the manual labeling and GrainID methods had comparably great performance in terms of both $Err_{i, median}$ and $V_{i, 3rd-1st}$.

### 4.2 Comparison of GrainID, BASEGRAIN and Wolman in diverse environments

The entire test dataset (Table 1) was used to evaluate the performance of GrainID, BASEGRAIN and Wolman methods in diverse fluvial environments. In Fig. 5, we present photos (Fig. 5a, 5e, 5i, 5m), and the predictive results of manual labeling (Fig. 4b, 4f, 4j, 4n), GrainID (Fig. 5c, 5g, 5k, 5o) and BASEGRAIN (Fig. 5d, 5h, 5l, 5p). The photos cover a variety of environments in which Fig. 5a is a flume sandy-gravel bed, Fig. 5e shows a flume bed with a wide grain size range and with inter-granular noise, Fig. 5i shows a forested riverbed with vegetation debris (from a small mountain watershed) and Fig. 5m is a drone photo of a large mountain riverbank.

A rough comparison shows that GrainID successfully predicts grains with inter-granular noise (Fig. 5g), while BASEGRAIN falsely recognizes the inter-granular noise as grain boundaries and splits those grains into smaller ones (Fig. 5h). When there is vegetation, GrainID distinguishes grains from large wood elements (Fig. 5k) while vegetation debris is frequently falsely predicted as grains by BASEGRAIN (Fig. 5l). For Fig. 5m, even with water in the image leading to some predictive error due to limited training for this uncalibrated site, GrainID performs well for all grain size groups (Fig. 5o). With BASEGRAIN, the error due to water was partly overcome ascribe to human expertise during the parameter tuning process, but the model falsely merges some small grains in the images (Fig. 5p).



As shown in Table 3, for small grains $D_5$, $D_{16}$, $D_{50}$, GrainID outperforms Wolman and significantly outperforms BASEGRAIN in terms of both $Err_{i, mean}$ and $Err_{i, median}$. For $D_{84}$ and $D_{95}$, GrainID and Wolman show similar performance while BASEGRAIN shows slightly lower performance than the other two methods. As for prediction variation (Fig. 6b), BASEGRAIN shows significantly larger variation $V_{i, 3rd-1st}$ than the GrainID and Wolman methods for all grain percentiles.

When comparing the change of predictive error versus grain percentiles, Wolman and BASEGRAIN both show larger predictive error for small grains than for large grains. In contrast, GrainID shows similar consistent performance for all grain percentiles. The results indicate that GrainID is a more accurate and robust grain sizing method (especially for small grains) than BASEGRAIN and Wolman methods for diverse fluvial environments.

### 4.3 Performance of GrainID in uncalibrated sites with airborne photos

To test the predictive ability of GrainID in uncalibrated rivers, 13 drone photos were compiled (Table 1). As shown in Table 3, GrainID shows slightly lower performance for all grain percentiles than its performance in diverse environments, where most of the evaluated images (53 out of 66) were from calibrated sites. Inversely, BASEGRAIN shows slightly higher performance in these conditions in comparison to its performance in diverse environments whilst the predictive error for Wolman in these rivers was similar to its predictive error in diverse environments. Once again, BASEGRAIN and Wolman consistently underestimate grain size (Fig. 6c), and show similar overall performance in terms of $Err_{i, mean}$ and $Err_{i, median}$. GrainID shows evidently outperform the two methods for all grain percentiles. As for prediction variation (Fig. 6c), GrainID and Wolman show similar variation in terms of $V_{i, 3rd-1st}$, and BASEGRAIN shows larger variation than the other two methods. The results suggest GrainID shows better predictive ability than BASEGRAIN and Wolman method even in uncalibrated rivers.

### 4.4 Influence of vegetation and inter-granular noise

The datasets were grouped based on the presence of vegetation and inter-granular noise in the image (Table 1) to evaluate the influence of vegetation and inter-granular noise on the GrainID, BASEGRAIN and Wolman methods. As shown in Table 3, the existence of vegetation and noise have little influence on the performance of GrainID in terms of both $Err_{i, mean}$ and $Err_{i, median}$ for all grain sizes. Conversely, BASEGRAIN shows larger $Err_{i, mean}$, $Err_{i, median}$ (Table 3) and prediction variation (Fig. 7b) for environments with vegetation and inter-granular noise. For vegetated environments, BASEGRAIN consistently shows larger $Err_{i, median}$ and $V_{i, 3rd-1st}$ for all $D_i$ compared to its performance in environments devoid of vegetation (Fig. 7b). For environments without the presence of inter-granular noise, BASEGRAIN consistently overestimates grain size for all $D_i$. Interestingly enough however, when there is inter-granular noise, BASEGRAIN consistently underestimates grain size for all $D_i$ (Fig. 7b). The performances of Wolman in the four test subsets in this section were similar for all grain percentiles, where there is limited influence from vegetation and inter-granular noise on the performance of the Wolman method (Fig. 7c). Overall, GrainID showed the smallest $Err_{i, median}$ and $V_{i, 3rd-1st}$, while BASEGRAIN showed the largest $Err_{i, median}$ and $V_{i, 3rd-1s}$ for environments with vegetation and inter-granular noise.



## 5 Discussion

In this section, we first discussed the error sources of different image-based methods based on the results in section 4. Subsequently, we explored the influence of image tile size and image resolution on the predictive ability of GrainID by varying

the image tile size and image resolution. Then, the truncation area for the smallest detectable grains was discussed and the model efficiency of different image-based methods was compared. Finally, the limitations of GrainID and future improvements and studies were discussed.

### 5.1 Error analysis

The error sources for image-based grain size measurement methods can be divided into five types: (1) the intrinsic error arising

from estimating three-dimensional grains with their projection on a two-dimensional image; (2) errors associated with the image-processing algorithm; (3) errors associated with sub-optimal environments from vegetation, inter-granular noise and sub-optimal lighting; (4) errors associated with image tile size and image resolution; and (5) errors associated with grain size distribution, irregular grain shape and photo distortion. Amongst the errors above, error type 1 is present for all image-based methods and has been widely discussed in previous literature (Graham et al., 2010) whilst error type 5 is likely to have little

influence on the final prediction results (Sime and Ferguson, 2003; Graham et al., 2005b; Detert and Weitbrecht, 2012). In this section, we will discuss the advantages and disadvantages of manual labeling, GrainID, BASEGRAIN and Wolman methods (error type 2), and discuss how vegetation, inter-granular noise, image tile size and image resolution influence the model's predictive performance (error type 3, 4).

Manual labeling, based on the operator's cognitive ability of identifying the grains is the most robust and reliable method when

applied to diverse fluvial environments. The influence of image resolution and image tile size on manual labeling are reduced compared to other models. However, the method is extremely time-consuming and laborious. In addition, the method requires a significant degree of expertise from the operator to correctly identify grains. Moreover, labeling error variates from operator to operator (Fig. 3), and usually increases (especially for fine grains) with operator fatigue when processing hundreds and thousands of grains. Manual labeling has been widely used as a baseline method for grain detection studies (Sime and Ferguson,

2003; Graham et al., 2005a; Ronneberger et al., 2015) and was used for the training and evaluation of models in this study.

The Wolman Pebble count is a semi-automatic grain size measurement method as it requires a manual measurement of at least 100 grains and as a result takes more time to perform in comparison to BASEGRAIN and GrainID. Wolman method shows consistent predicting ability in diverse environments. Vegetation, inter-granular noise, sub-optimal lighting and image resolution have similar influence on the method (Table 3) as seen in Manual labeling methods. However, the predicting ability

of Wolman method is sensitive to grain size distribution. The Wolman method shows better predicting ability for large grains ($D_{84}$, $D_{95}$) than small grains ($D_5$, $D_{16}$, $D_{50}$; Table 3), and the method is limited to material > 8 mm when applied in mountain rivers (Kellerhals and Bray, 1971).



BASEGRAIN, as an automatic grain-detecting model, is less time-consuming than manual labeling and Wolman method and is capable of measuring the spatial distribution of grains. The method has been proven in studies to be a reliable grain size measurement method under optimal conditions (no inter-granular noise, no vegetation, and uniform lighting and dryness) (Detert and Weitbrecht, 2020). For flume experiments with regular sandy-gravel beds, BASEGRAIN shows good performance for predicting large grains when compared to sieving results (Fig. 5b). However, as shown in Fig. 4 and Fig. 5, the model performs poorly in detecting very fine grains (usually less than 50 pixels) even in environments with optimal conditions. In addition, the performance of BASEGRAIN in predicting large grains was highly sensitive to environmental factors such as vegetation and inter-granular noise. BASEGRAIN had poor and inconsistent performance for sub-optimal environments (Table 3), while the model also evidently overestimates grains without inter-granular noise while underestimating grains with inter-granular noise (Fig. 7b). The reasons are as follows: as shown in Fig.4, although BASEGRAIN applied a well-designed interstice-detecting algorithm, the model is based on detecting grain boundaries. When there is vegetation or inter-granular noise in the image, the BASEGRAIN algorithm intrinsically falsely detects the edges of vegetation or inter-granular noise as the edges of grains. Moreover, as shown in section 4.1, due to the limitations of image resolution, the boundaries of small grains are unclear and detected poorly with simple thresholds. In addition, the model contains 46 adjustable parameters (in which seven are key parameters) such that BASEGRAIN requires a sophisticated parameter tuning process and a high level of expertise from the operator when applied to suboptimal conditions such as field images.

GrainID, based on CNN, not only uses grain interstice information but also high-level grain features such as shape, color or texture to make their final predictions (Buscombe, 2020). The analysis in section 4.2 showed GrainID evidently outperforms the Wolman and BASEGRAIN for all grain percentiles for diverse environments, the advantage of GrainID is more significant for small grains than for large grains (Table 3). For a flume experiment with a sandy-gravel bed, GrainID showed predictive ability comparable to baseline sieving results with an $Err_{5, median} = 0.16$, $Err_{16, median} = 0.16$, and $Err_{50, median} = 0.16$ (similar to $Err_{i, median} = 0.16$, $Err_{16, median} = 0.10$, and $Err_{50, median} = 0.15$ of Manual labeling). In addition, the architecture (Fig. 4) of GrainID overcomes errors arising from image splits (poorer predicting ability of CNN at the border region of an image tile), making it a promising method for large-scale drone surveys. The analysis on drone photos in section 4.3 showed the potential of applying GrainID in large-scale river survey. Similar to other machine learning methods, the predictive ability of GrainID is highly dependent on the quality of training datasets such as the number and diversity of training images. In section 5.5, we discussed the limitations of GrainID and the issue of lack of training in detail.

## 5.2 Influence of image tile size and resolution

The model's predictive ability will be influenced by whether the size of image tiles are too large (under-split; limited by the GPU memory) (Ronneberger et al., 2015) or small (over-split; limited by the size of largest grain to detect). Based on the forested mountain river and sparsely vegetated large river datasets (Table 1), we explored the influence of image tile size on grain detection ability by varying the image tile size (64*64, 128*128, 256*256, 512*512, 768*768, 1024*1024) while maintaining the raw image resolution. As shown in Fig. 8a, the tile size 64*64 yielded positive predictive results for small



grains ($D_5$, $D_{16}$, $D_{50}$) while it failed to detect larger grain classes ($D_{84}$, $D_{95}$). The tile sizes 128*128, 256*256 and 512*512 had a similar predictive accuracy for all grain size percentiles, with 512*512 showing the lowest $Err_{i,mean}$ for $D_{50}$, $D_{84}$, $D_{95}$ and the lowest average $Err_{i,mean}$ (average $Err_{i,mean}$ is the averaged value of $Err_{i,mean}$ for all grain percentiles). The average $Err_{i,mean}$ increases with increasing tile size for tile sizes larger than 512*512.

Based on the SAFL dataset in which manual sieving data was collected (Table 1), we explored the influence of image resolution on grain size detection by down-sampling the original image resolution of 0.45 mm/pixel up to 4.5 mm/pixel and comparing the results of down-sampled images to the sieving results. The down-sampling was done using a simple moving average method of increasing window size from 1*1 up to 10*10 (the later controls the spatial resolution) (Chen et al., 2020). As shown in Fig. 8b, the predictive error was quite consistent ($Err_{i,mean} \sim 0.10$) for resolutions higher than 1.8 mm/pixel, and increased
slowly (from 0.10 to 0.96) for resolutions from 1.8 mm/pixel to 3.15 mm/pixel and sharply for resolutions greater than 3.15 mm/pixel. $Err_{i,mean}$ for small grains were more sensitive to the variable of image resolution than large grains. The analysis showed that for a sandy-gravel bed with $D_{50}$ = 9.5 mm, GrainID can predict all grain percentiles for image resolutions higher than 1.8 mm but failed to predict grain sizes for resolutions lower than 3.15mm/pixel.

**5.3 Smallest detectable grains**

The ability to detect fine grains is limited by image resolution for all image-based grain sizing algorithms. For the smallest detectable grains, *Graham et al.* (2005a, 2005b) proposed that the measurement error increases sharply for grains with a b-axis smaller than 23 pixels, while *Detert and Weitbrecht* (2012, 2020) adopted a grain area of 23 pixels as the lowest truncation value (the area of smallest detectable grains) to detect grains for BASEGRAIN. Based on the SAFL dataset, we calculated the predictive error $Err_{i,mean}$ of GrainID in comparison to sieving results for different grain area truncation values ($area_{trunc}$). As
shown in Fig. 9, the predictive error of $D_5$ is very sensitive to $area_{trunc}$, $Err_{5,mean}$ slowly decreases from 0.22 to 0.19 for increasing $area_{trunc}$ from 1 to 18 pixels, had the lowest value of 0.19 for $area_{trunc}$ between 18 – 25 pixels, and sharply increases to 0.53 for increasing $area_{trunc}$ from 25 to 100 pixels. The $Err_{16,mean}$, $Err_{50,mean}$ and $Err_{84,mean}$ are less sensitive to $area_{trunc}$ compared to $Err_{5,mean}$. However, they have similar three-stage trends to increasing $area_{trunc}$, where the error values first decrease with increasing $area_{trunc}$ (stage-1), then reach a minimum value for an $area_{trunc}$ period (stage-2), and finally increase for
increasing $area_{trunc}$ (stage-3). In stage-1, the negative correlation between $Err_{i,mean}$ and $area_{trunc}$ suggests that the smallest detectable grain for GrainID are grains with an area of 18 pixels. In stage 3, the positive correlation between $Err_{i,mean}$ and $area_{trunc}$ suggests that the $area_{trunc}$ is too large so that the correct predictions of GrainID were wrongly filtered out. For $D_{95}$, similar to the previous findings (Graham et al., 2005a), the result shows that $Err_{95,mean}$ are unaffected by $area_{trunc}$ and remain almost constant for $area_{trunc}$ from 1 to 100. The analysis above suggests that GrainID performs optimally when the grain area
truncation values were equal to 18 - 25 pixels.



## 5.4 Model efficiency

To compare the efficiency of GrainID, BASEGRAIN and Wolman methods, we calculated the time consumed by the three models for predicting images from three typical environments (Table 1): (1) SAFL datasets: 26 images from flume experiments with optimal conditions (Singh et al., 2013); (2) MCHEL datasets: 12 images from flume experiments with sediment with inter-granular noise (Wang et al., 2021) and (3) 15 images from forested mountain rivers (Brayshaw, 2012). For GrainID, BASEGRAIN and Wolman methods, the rough averaged time of processing an image are 5s, 46s and 962s for SAFL datasets; 21s, 300s and 1000s for MCHEL datasets and 22s, 600s and 1000s for the forested rivers datasets (processing time of GrainID depends on GPU, our GPU is GTX 1080Ti).

GrainID, working autonomously, had the highest efficiency in terms of the time required to analyze the images. Wolman method had a significantly lower working efficiency as the model necessitated the Manual labelling of the 100 sampled grains. The working efficiency of BASEGRAIN varies in different application environments, BASEGRAIN necessitated much more time for parameter tuning for images from MCHEL and forested rivers than images from SAFL. Images from MCHEL and forested rivers contained significantly different types of images and as such to implement the use of BASEGRAIN required an arduous parameter tuning process and a significant level of expertise.

## 5.5 Limitations and future work

We tested the robustness and applicability of GrainID by applying it to uncalibrated sites (section 4.3). As our model was trained by more than 65,000 grains under diverse mountain environments, the method was overall robust and outperformed BASEGRAIN and Wolman even for uncalibrated sites. However, the test datasets of uncalibrated sites only included 13 images from four sparsely vegetated mountain rivers. As shown in Fig. 5, due to a lack of training some large wood debris, unresolved cohesive sands, flow wave and drone marker boards were falsely identified as grains by the program. As such, the application of GrainID to more diverse fluvial environments would require more training datasets from a greater variety of environments. However, preparing training datasets necessitates the use of manual labeling and is therefore time-consuming and laborious. For some images with dense vegetation, even experienced operators may have trouble confidently identifying grains (especially small grains) in the images. Meanwhile, as seen in many other object-based methods, the smallest grain size identifiable by GrainID is limited by image resolution and the grain pattern learned by the model is limited by image tile size. In addition, the present model only identifies the presence of sediment grains in the image in which they were further segmented into pixels either as grains or interstices. We hope that with further development the model can be applied to a greater variety of environments and can identify vegetation, cohesive sand or other environmental elements so that the model can learn to further distinguish different environmental elements in the image.

For model efficiency, GrainID requires much less time in implementation in comparison to the BASEGRAIN and Wolman methods. However, with the help of parallel computing, there are already successful real-time image segmentation techniques in commercial use such as the introduction of self-driving cars and robotic perception (Treml et al., 2016; Siam et al., 2018).



With more studies on improving the accuracy and efficiency of GrainID, the model could be applied to detect grains in video
recordings of flume experiments which is very important for studies on sediment mobility and transport in gravel-bed rivers.

Meanwhile, our study indicates that GrainID has the potential to be used towards predicting drone photos. With more studies
on applying GrainID to drone images, the model could be applied to watershed-scale surveys to study the changes and spatial
distribution of grain sizes in a watershed.

## 6 Conclusion

We proposed an image-based grain detecting model (GrainID) based on convolutional neural networks to detect sediment

grain size in diverse fluvial environments. To develop the model, we compiled a dataset of 84 flume and 118 field photos
containing more than 115,000 grains covering environments under a wide range of vegetation coverage, grain lithology and
lighting conditions.

Tests were performed to compare the predictive ability of GrainID with the performance of manual sieving, manual labeling,
BASEGRAIN and Wolman pebble count methods. When using manual sieving as a baseline result, for a flume experiment

with sandy-gravel bel, GrainID, with $Err_{i, median}$ = 0.16, 0.16, 0.16, 0.23 and 0.24 for $D_5$, $D_{16}$, $D_{50}$, $D_{84}$, $D_{95}$, showed a predictive
ability comparable to manual labeling ($Err_{i, median}$ = 0.16, 0.10, 0.15, 0.14 and 0.15 respectively) especially for smaller grains.
GrainID and manual labeling largely outperform BASEGRAIN and Wolman method for smaller grains ($D_5$, $D_{16}$, $D_{50}$), but
show similar performance with BASEGRAIN and the Wolman method for larger grains ($D_{84}$, $D_{95}$).

For the entire test dataset based on a diverse range of environments, when using manual labeling as the baseline result, GrainID

showed the overall best performance and maintained its advantage even in uncalibrated rivers, whereas BASEGRAIN showed
the overall worst performance. The test datasets were grouped based on the presence of vegetation and inter-granular noise in
the image (Table 1) to evaluate the influence of vegetation and inter-granular noise on the three image-based methods. The
results showed that vegetation and inter-granular noise have little influence on the predictive ability of GrainID and Wolman
methods, while BASEGRAIN showed inconsistent predictive ability and larger $Err_{i, median}$ and $V_{i, 3rd-1st}$ in environments with

vegetation and inter-granular noise.

We also studied the influence of image tile size and resolution on the predictive ability of GrainID. For the forested mountain
rivers and sparsely vegetated large river datasets, GrainID with an image tile size = 512*512 pixel*pixel had the best
performance. For a sandy-gravel bed with $D_{50}$ = 9.5 mm, the GrainID performed optimally when the image resolution was
higher than 1.8 mm/pixel and the grain area truncation values (the area of smallest detectable grains) were equal to 18 - 25

pixels. The analysis also indicated that GrainID had a higher working efficiency than the BASEGRAIN and Wolman methods
in terms of processing time. The working efficiency of BASEGRAIN is sensitive to environmental conditions, whilst the
average efficiency of GrainID only depended on the size of the input images. Conversely, the average time for Wolman method
analysis was constant for different environments. The error sources of different methods were also discussed, and the



limitations and potential of GrainID for detecting sands and vegetation, as well as real-time prediction and watershed-scale
application deserve further studies and development.

*Code and data availability*. Data sets and GrainID model code available at https://zenodo.org/record/5240906

*Author contribution*. Xingyu Chen prepared the data, established the model, wrote the model code and produced the majority
of the paper. Marwan Hassan contributed significantly to the data, the model, the paper and the original idea of the work.
Xudong Fu provided essential help to the data, the original idea of the work and editorial feedback to improve the paper.

*Competing interests*. The authors declare that they have no conflict of interest.

*Acknowledgments*. Shawn Chartrand, Tobias Muller, Sam Anderson commented on the early work. Xingyu Chen, Cormac
Chui, Lily Liu, Yongpeng Lin, Kai Sun prepared the manual labels. Drew Brayshaw, Carina Helm provided field photos.
Jiamei Wang and Xingyu Chen conducted the flume experiment in the University of British Columbia. Cormac Chui
commented on the paper. Eric Leinberger prepared the figures. The visit to the University of British Columbia for Xingyu
Chen were supported by the China Scholarship Council (file NO. 201906210321). This study was funded by the Natural
Sciences and Engineering Research Council of Canada (NSERC) Discovery Grants (M. A. H. [RGPIN 249673-12]). The
participation of Xingyu Chen and Xudong Fu was supported by the National Natural Science Foundation of China (NSFC)
under Grant Nos. 91747207 and 51525901.

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





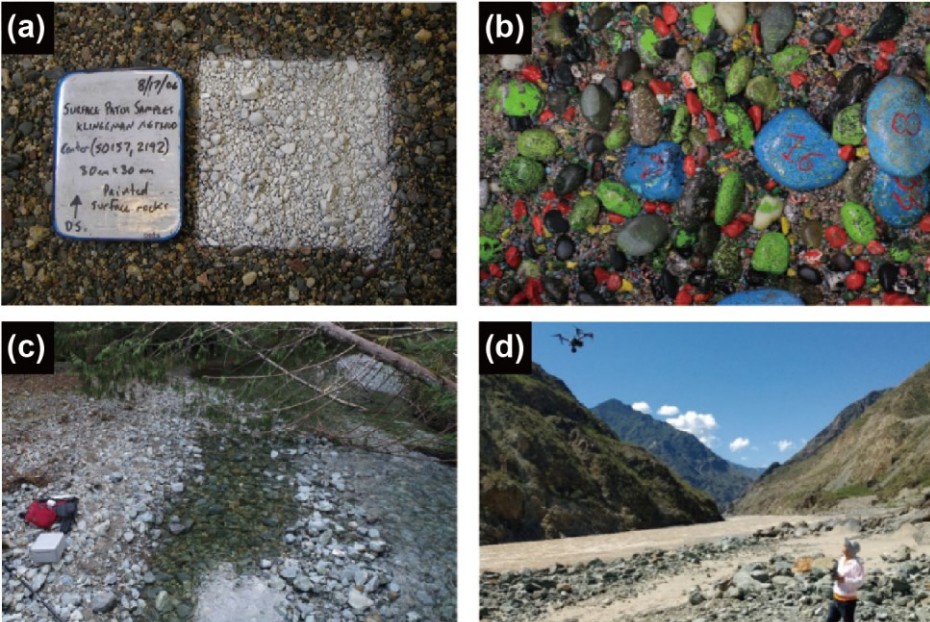


**Figure 1: Four typical environments in our datasets: (a) a bed sample collected in SAFL; (b) a step-pool channel bed in MCHEL; (3) Carnation Creek; (d) Upper Yangtze River.**





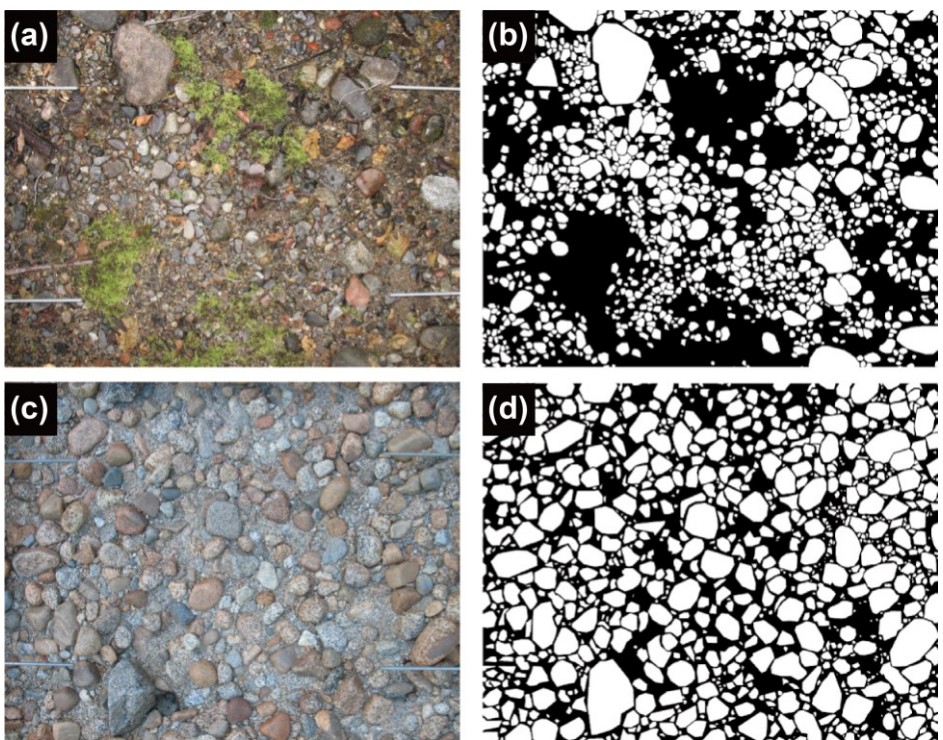

**Figure 2: Examples of two field photos and the corresponding manual labels: (a) a photo with vegetation from Anderson Creek and (b) the corresponding manual label; (c) a photo without vegetation from Coquitlam River and (d) the corresponding manual label.**



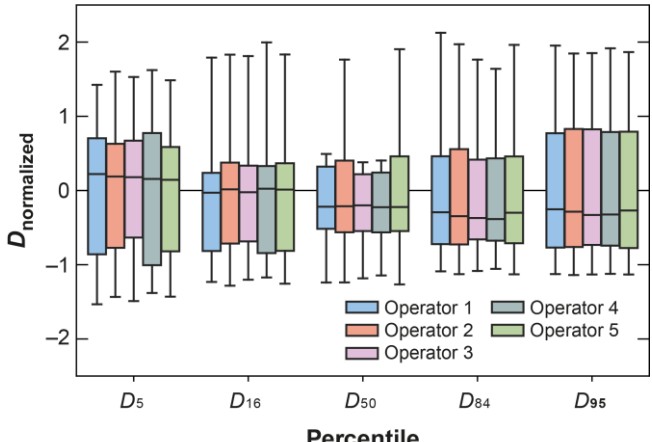

**Figure 3: Boxplot of normalized grain size $D_{normalized}$ for percentiles $D_i$ for five human labeling operators.**





**Figure 4: Framework and specific algorithms of GrainID: (a) GrainID framework; (b) border extrapolation and overlap-tile prediction; (c) image augmentation; and (d) u-net architecture adapted from Ronneberger et al. (2015).**

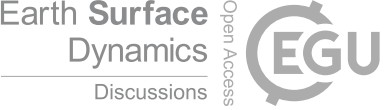



**Figure 5: Vertical photos and predicting results of Manual labeling, GrainID and BASEGRAIN for a variety of environments: (a-d) flume sandy-gravel bed (SAFL dataset); (e-h) flume gravel bed with inter-granular noise (MCHEL dataset); (i-l) location with dense vegetation (Sullivan Creek); (m-p) drone photo of an uncalibrated large river bank (Yangtze River).**



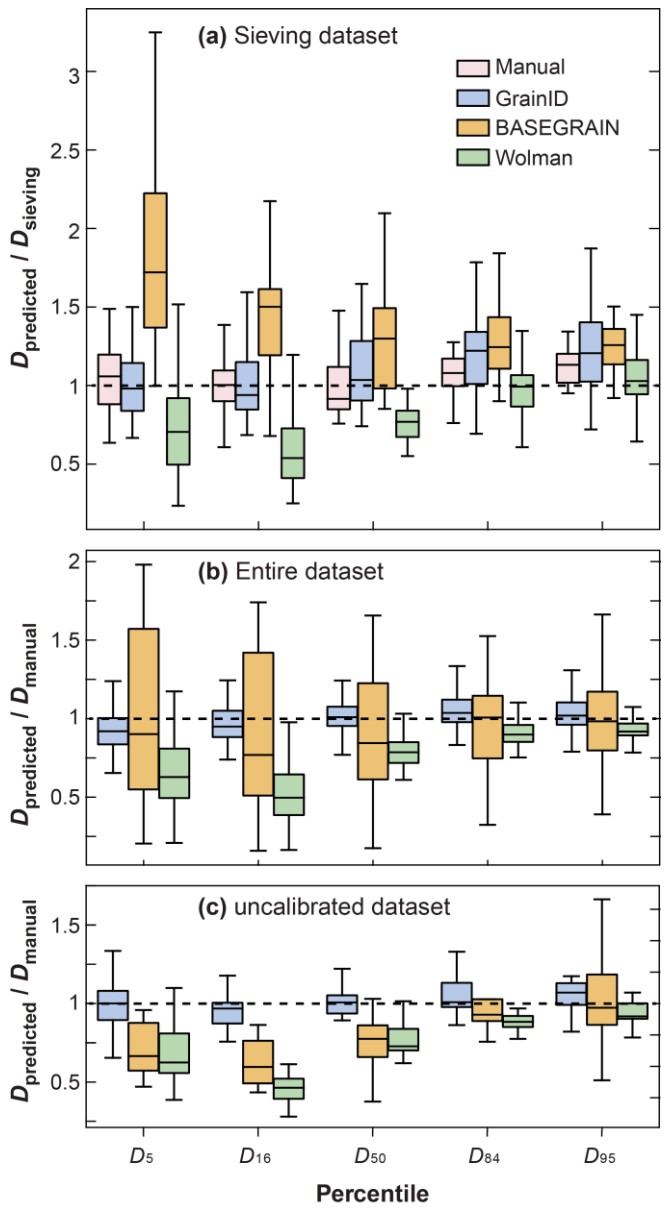

**Figure 6: Performance comparison for different methods. (a)** $D_{predicted}/D_{sieving}$ **shown for grain percentiles** $D_i$ **of Manual labeling, GrainID, BASEGRAIN and Wolman methods (referred to as G, B and W methods respectively) for a flume sandy-gravel bed; (b)** $D_{predicted}/D_{manual}$ **shown for** $D_i$ **of G, B and W methods for the entire datasets; (c)** $D_{predicted}/D_{manual}$ **shown for** $D_i$ **of G, B and W methods for uncalibrated rivers with drone photos.**



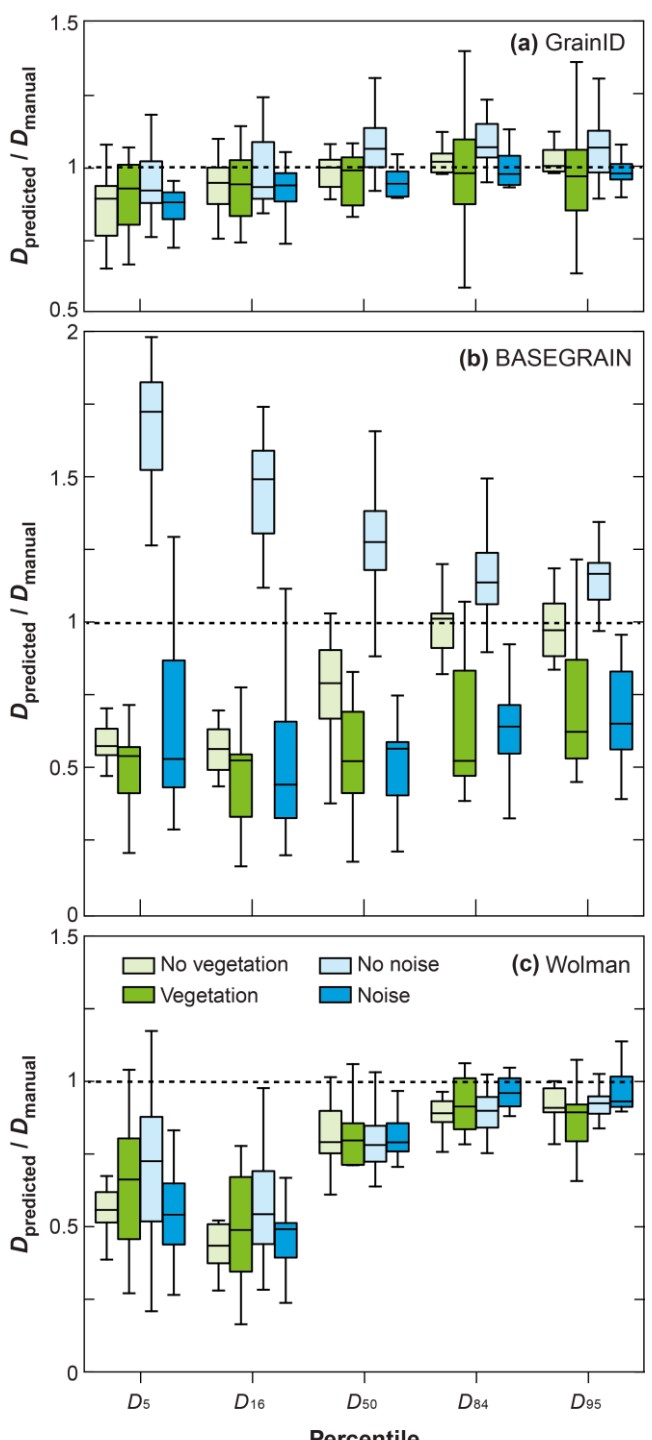

**Figure 7: Ratio of predicted to baseline grain size value shown for different $D_i$ for (a) GrainID, (b) BASEGRAIN and (c) Wolman method in environments with/without vegetation and inter-granular noise.**



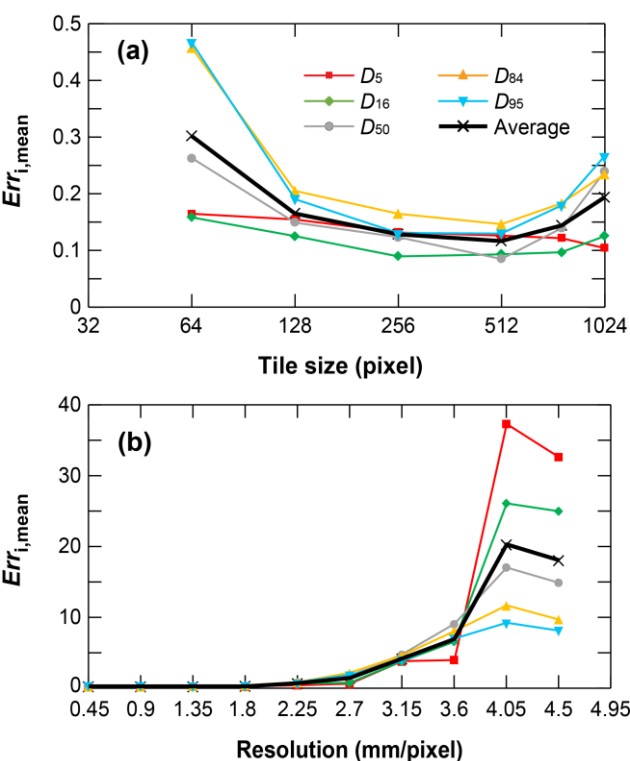

**Figure 8: (a) Prediction accuracy of different grain percentiles for (a) different image tile size; (b) different image resolution.**





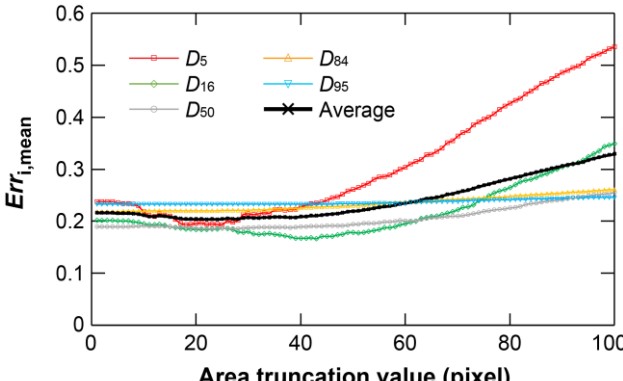

**Figure 9: Prediction error versus area truncation value of smallest detectable grains.**



**Table 1: Description of datasets**

| Stream/flume | Basin Area (km²) | Slope (%) | $D_{50}$ (mm) | $D_{84}$ (mm) | # of Trained Images | # of Tested Images | Image Resolution (mm/pixel) | Reference | Comments |
|---|---|---|---|---|---|---|---|---|---|
| **Flume** | | | | | | | | | |
| MCHEL, CA | | 6-8 | 15.0 | 30.0 | 21 | 12[i] | 0.1 mm | Wang et al. (2021) | sandy gravel with inter-granular noise. |
| SAFL, USA | | 0.3-1.6 | 9.5 | 15.5 | 25 | 26[ni] | ~0.4 mm | Singh et al. (2013) | sandy-gravel bed sampled with Klingeman protocol |
| **Field: Forested mountain river** | | | | | | | | | |
| Albert River | 69.7 | 0.8 | 22.1 | 40.7 | 3 | / | ~0.3 mm | | muddy; a lot of fines. |
| Arrow Creek | 78.7 | 2.8 | 51.7 | 110.4 | 3 | 1[v] | 0.3 mm | | covered by fallen conifer leaves |
| Cabin Creek | 93.2 | 1.7 | 77.3 | 176.0 | 4 | / | 0.3 mm | | wet; non-uniform lighting |
| Coquitlam River | 54.7 | 0.7 | 28.6 | 46.3 | 4 | 2[nv] | 0.3 mm | | porphyritic granite with inter-granular noise |
| East Creek | 1.21 | 1.6 | 44.6 | 82.1 | 7 | 1[v] 1[nv] | 0.3 mm | | wet; deciduous broad-leaved forest |
| Norris Creek | 79 | 3.1 | 69.7 | 186.0 | 3 | / | 0.3 mm | | sparsely vegetated |
| Split Creek | 81.3 | 3.6 | 28.3 | 79.0 | 3 | / | 0.3 mm | | metamorphic lithology; sparsely vegetated |
| Deer Creek | 80.5 | 2.6 | 56.2 | 124.2 | 2 | 1[v] 1[nv] | ~0.3 mm | | moss-covered porphyritic granite |
| Ambusten Creek | 32.9 | 6.8 | 14.8 | 32.3 | 3 | / | ~0.3 mm | | muddy; metamorphic lithology; moderately vegetated |
| Anderson Creek (Hat) | 31.9 | 6.9 | 54.0 | 186.71 | 2 | 1[v] | ~0.3 mm | Brayshaw (2012) | covered by fallen fine conifer leaves; moss-covered porphyritic granite |
| Fell Creek | 4.4 | 18.4 | 59.7 | 138. | 4 | 2[v] | ~0.3 mm | | granite; heavily vegetated |
| Hidden Creek | 56.7 | 4.4 | 118.0 | 236.8 | 1 | 1[v] | ~0.3 mm | | Intrusive and extrusive lithologies |
| Hosmer Creek | 6.4 | 8.5 | 38.8 | 113.0 | 2 | / | ~0.3 mm | | moss-covered granite; non-uniform lighting |
| Kanaka Creek | 47.7 | 1.0 | 89.0 | 195.9 | 1 | 1[v] | ~0.3 mm | | granite covered by heavy moss |
| Noons Creek | 1.6 | 6.0 | 39.0 | 88. | 2 | 2[v] | ~0.3 mm | | wet; muddy granite; covered by fallen conifer leaves |
| Redfish Creek | 26.2 | 7.2 | 80.3 | 163.2 | 2 | / | ~0.3 mm | | porphyritic granite covered by moss heavy moss and fallen conifer leaves |
| Sullivan Creek | 6.22 | 17.0 | 40.7 | 99.40 | 2 | 1[v] | ~0.3 mm | | granite covered by plant branches and fallen conifer leaves |
| Carnation Creek, BC, CA | 11.2 | 1.6 | 23.3 | 48.3 | / | 7[nv] | 0.5 mm | Helm et al. (2020) | Drone photos; non-uniform lighting; sparsely vegetated |
| **Field: Typical large rivers** | | | | | | | | | |
| Yangtze River | >100,000 | ~0.1 | | | / | 3 | ~10 mm | This study | Drone photos; non-uniform lighting; sparsely vegetated |
| Yaluzangbu River, China | >100,000 | ~0.6 | | | / | 2 | ~10 mm | This study | |
| Yaluzangbu River Tributary, China | >1000 | ~0.6 | | | / | 1 | ~10 mm | This study | Drone photos; wet cohesive bed; non-uniform lighting |



**Table 2: Description of each image processing step of GrainID**

| Procedures | Operation | Description |
|---|---|---|
| *Step 1:* *Pre- processing* | 1.1 - image extrapolation-1 | If the size (e.g. 2000*2000) of original input image can't be equally split into multiple 512*512 tiles, the image was extrapolated into 2048*2048 based on mirroring the right and down image border region. |
| | 1.2 - image extrapolation-2 | Based on the overlap tiles strategy, for prediction of image border region, the missing context was extrapolated by mirroring the border region. |
| | 1.3 - contrast filter | A Sigmod contrast filter in Python Library *pillow* was applied. |
| | 1.4 - image augmentation | The input images were augmented by applying 0º, 90º and 180º counter-clockwise (CCW) rotation and horizontal and vertical flip. |
| | 1.5 - image split | Input images were split into overlapping image tiles (512*512) as dashed red and blue rectangles in Fig 4b. |
| *Step 2:* *Prediction* | 2.1 - *U-Net* prediction | All image tiles were then sequentially input into U-Net for prediction. |
| | 2.2 - recombination | The predicted image tiles were recombined into a full image. |
| | 2.3 – assemble vote | The five predictions from augmented images vote for the assemble result. |
| *Step 3:* *Post- processing* | 3.1 - filling holes | The holes inside grains were filled. |
| | 3.2 - filter fine grain | Unresolvable grains with size < 20 pixels were deleted. |
| | 3.3- narrowing interstice | An inverse watershed algorithm was applied. |
| | 3.4 - watershed algorithm | A watershed algorithm was performed for further separation. |





**Table 3: Median and mean predicting error for different grain zizing methods and for different evaluating datasets with manual as baseline method.**

| Datasets | Percentile | GrainID | | BASEGRAIN | | Wolman | |
|---|---|---|---|---|---|---|---|
| | | $Err_{mean}$ | $Err_{median}$ | $Err_{mean}$ | $Err_{median}$ | $Err_{mean}$ | $Err_{median}$ |
| Entire datasets | $D_5$ | 0.13 | 0.11 | 0.50 | 0.50 | 0.36 | 0.37 |
| | $D_{16}$ | 0.10 | 0.10 | 0.46 | 0.47 | 0.49 | 0.50 |
| | $D_{50}$ | 0.10 | 0.06 | 0.35 | 0.33 | 0.23 | 0.21 |
| | $D_{84}$ | 0.12 | 0.07 | 0.25 | 0.20 | 0.13 | 0.11 |
| | $D_{95}$ | 0.12 | 0.08 | 0.24 | 0.18 | 0.11 | 0.09 |
| Uncalibrated Sites for GrainID | $D_5$ | 0.15 | 0.11 | 0.32 | 0.36 | 0.33 | 0.38 |
| | $D_{16}$ | 0.11 | 0.11 | 0.38 | 0.40 | 0.50 | 0.54 |
| | $D_{50}$ | 0.12 | 0.06 | 0.28 | 0.24 | 0.27 | 0.27 |
| | $D_{84}$ | 0.15 | 0.07 | 0.20 | 0.11 | 0.12 | 0.12 |
| | $D_{95}$ | 0.17 | 0.13 | 0.23 | 0.16 | 0.12 | 0.08 |
| Datasets with vegetation | $D_5$ | 0.13 | 0.07 | 0.48 | 0.46 | 0.36 | 0.34 |
| | $D_{16}$ | 0.11 | 0.10 | 0.51 | 0.48 | 0.52 | 0.51 |
| | $D_{50}$ | 0.10 | 0.05 | 0.46 | 0.48 | 0.27 | 0.20 |
| | $D_{84}$ | 0.18 | 0.10 | 0.36 | 0.48 | 0.21 | 0.13 |
| | $D_{95}$ | 0.17 | 0.08 | 0.33 | 0.38 | 0.17 | 0.14 |
| Datasets without vegetation | $D_5$ | 0.15 | 0.11 | 0.41 | 0.43 | 0.43 | 0.44 |
| | $D_{16}$ | 0.08 | 0.08 | 0.44 | 0.44 | 0.51 | 0.57 |
| | $D_{50}$ | 0.07 | 0.05 | 0.23 | 0.21 | 0.19 | 0.21 |
| | $D_{84}$ | 0.07 | 0.03 | 0.11 | 0.07 | 0.12 | 0.11 |
| | $D_{95}$ | 0.06 | 0.05 | 0.10 | 0.10 | 0.11 | 0.09 |
| Datasets with inter-granular noise | $D_5$ | 0.13 | 0.12 | 0.40 | 0.47 | 0.46 | 0.46 |
| | $D_{16}$ | 0.09 | 0.06 | 0.51 | 0.56 | 0.55 | 0.51 |
| | $D_{50}$ | 0.09 | 0.05 | 0.50 | 0.44 | 0.19 | 0.21 |
| | $D_{84}$ | 0.07 | 0.06 | 0.38 | 0.36 | 0.11 | 0.07 |
| | $D_{95}$ | 0.06 | 0.04 | 0.33 | 0.35 | 0.08 | 0.08 |
| Datasets without inter-granular noise | $D_5$ | 0.11 | 0.10 | 0.67 | 0.72 | 0.31 | 0.27 |
| | $D_{16}$ | 0.12 | 0.10 | 0.46 | 0.49 | 0.43 | 0.46 |
| | $D_{50}$ | 0.10 | 0.07 | 0.30 | 0.28 | 0.22 | 0.22 |
| | $D_{84}$ | 0.11 | 0.08 | 0.18 | 0.14 | 0.12 | 0.10 |
| | $D_{95}$ | 0.11 | 0.08 | 0.18 | 0.17 | 0.09 | 0.08 |