# Peer review of "CNN for image-based sediment detection applied to a large terrestrial and airborne dataset"

_Earth Surface Dynamics, 2021_

## Author Response (AR1)

**Response file to comments**

**1. Response to referee #1, Byungho Kang**

**Byungho Kang:**
The paper explores the applicability of CNN-based image segmentation on determining the grain size, which could eventually substitute the pre-existing methods for grain detection. The authors verify the reliability of the proposed grain sizing method by comparing it with hand-measured labels and manual sieving.

Promising as it is, the paper needs a small number of revisions.
**Response:**
Thank you for the nice summary of our work.

**Byungho Kang:**
It seems necessary to elaborate the general training process for the U-net (e.g., hyperparameters values, number of epochs for training, what kind of optimization method were used, how the tiles were being selected for training, et Cetra). The inclusion of a section or a table could help.
**Response:**
We added a paragraph (L201-L209) to elaborate upon the U-net implementation.
'The U-net was implemented based on a python library *pytorch* (Paszke et al., 2019). The cross entropy loss function and the stochastic gradient descent were used for model optimization. Model hyperparameters were tuned based on grid searching optimization and 5-fold random cross validation (Goodfellow et al., 2016). The training speed for *U-net* is influenced by the number of images in the training datasets, the batch size and the number of training epoch. Given a fixed training datasets, the hyperparameter *number of training epoch* was tuned first, followed by the *learning rate*. The maximum *batch size* depends on GPU memory and we preferred larger *batch size* for faster training speed when several *batch size* values result in a similar error during the cross-validation. The optimized model hyperparameters are: (1) *number of training epoch* = 150; (2) *learning rate* = 0.005; (3) *batch size* = 96; and (4) *image tile size* = 512. The optimum image tile size was determined based on the analysis in section 5.2. '

**Byungho Kang:**
Likewise, it would be better to provide more information on manual labeling. Even labeled by two observers, labeling 128,461 grains would take significant hours, and I assume it was based on an auto/semi-automatic algorithm that captures the grains from images.
**Response:**
To better train and evaluate our method, the 128,461 grains manual labeling dataset were all prepared by hand. The two observers cross-checked their results with each other, such that all the manually labeled images were from the consensus of two observers.
We added more details in the preparation of the manual labeling process in the manuscript (L137-L142).

**Byungho Kang:**

It would be better to include information about the number of grains in test images (both for manual labeling and U-net based prediction). This could improve the overall credibility (of error calculation) than merely providing the percentile-based information.

**Response:**

Accepted. We added a column 'Average # of Grains in each image' in Table 1 for manual labeling. Since manual labeling was used as the baseline method in Section 4.2, 4.3 and 4.4, we think that the number of grains for manual labeling is sufficient. Meanwhile, the percentile-based error is based on the calculation of grain weight and can provide different information from the number of grains in each image.

**Byungho Kang:**

The comparison between the sieving dataset and other image-based methods(Fig.6a) needs more clarification (e.g., conversion of physical scale (mm to pixels) should be mentioned in section 4, not 5)

**Response:**

Revised. We added a paragraph (L248-L251) to better explain the comparison between sieving and other methods (e.g., the conversion of physical scale and the correction of image photo distortion). We also added a section to introduce how a predicted image was transferred to grain size information (L187-L192).

We implemented a systematic analysis on how image resolution influences the predictive ability of Grain ID in section 5.3, the topic is different with the comparison of sieving, we think the section should be in the discussion part not in the result part.

**Byungho Kang:**

Some minor typos at line 244: Fig. 4b should be replaced with Fig. 5b.

**Response:**

Revised.

**2. Response to referee #2, anonymous referee**

**Referee:**
The authors develop a new deep learning method to determine grain size from photographs. This is done by assembling a dataset of images of grain size, manually labeling the images, and developing an analysis pipeline that includes basic computer vision as well as a fully convolutional neural network. They compare the results to manual counts, as well as an existing computer vision technique for grain size determination from pictures.

**Response:**
Thank you for the nice summary of our work.

My comments are as follows:

**Referee:**
Abstract: I would cite BASEGRAIN if it is discussed multiple times in the abstract.

**Response:**
Agreed. We cited BASEGRAIN in the abstract.

**Referee:**
L9: 'current methods are largely based on detecting grain intersticies". In the paper you mention many techniques that do not use this technique, so is this true?

**Response:**
Has been revised. Current image-based automated grain sizing methods can be classified from percentile-based to object-based methods. Only those object-based methods rely on detecting grain interstices. We rephrased the sentence and made it more accurate (L9).

**Referee:**
L21: I would go with 512 pixel *512 pixel or 512 *512 pixels

**Response:**
Accepted, we revised it to 512 *512 pixels.

**Referee:**
L25 'is time consuming'

**Response:**
Has been corrected.

**Referee:**
L41: I think Rubin should be mentioned here.

**Response:**
Accepted.

**Referee:**
L70: 'CNN techniques have proven..'

**Response:**

Has been corrected.

**Referee:**
L71: what are 'suboptimal conditioned' images?
**Response:**
Has been revised.
'… when applied to sub-optimally conditioned images (e.g. non-uniform lighting, noise due to organic debris).'

**Referee:**
L72: My understanding is that the classic Ronneberger UNET is not considered 'state of the art' at this point, considering how fast the field is moving.
**Response:**
Agreed. We corrected the sentence to 'one of the most widely-used CNN architecture …'

**Referee:**
L77-L86: this section should be rewritten for clarity. I got a bit lost. What did Mueller do? And where has the UNET been applied to orthophotographs?
**Response:**
We rewrote the section L75-L85 to make it more readable.

**Referee:**
L119 and Table 1: First, I do not count 136 images in the training column of table 1. can you explain? Second, is this the train/validation split? Or the train+validation/test split? How much data was used for training, validation, and testing? Was there a true hold out test set that was not looked at until the model was fully trained?
**Response:**
For the first question, to train the machine learning model to better distinguish sediments from field environmental elements and improve the model robustness, we specifically collected 42 field photos primarily consisting of various environmental elements (e.g. cohesive sands, wood, vegetation and water) with limited sediment grains in the images in addition to the 94 images in the training column of Table 1. In the revised manuscript, we added these 42 images to the training column of table 1, and we added a section to explain why we used these 42 photos to train the model (L95-L98).
For the second question: This is the 'train+validation/test', the test subset of 66 images is a true hold out set that was not looked at until the model was fully trained. Indeed, we realized we should introduce the cross-validation more clearly, so we added more details for the split of the dataset ('train+validation/test') at L119-L122, and for the cross-validation methods (L200-L203).

**Referee:**
L126-128: please revise for clarity. why is Ronneberger cited here when you are discussing grain size studies?
**Response:**
Has been revised. We deleted the Ronneberger citation.

**Referee:**

L135: 203 or 202?

**Response:**

Has been revised. It should be 202.

**Referee:**

L135, 138: remove word 'masters' from here and any other place in the ms. Replace with something else

**Response:**

Has been revised. We replaced all instances of the word 'master' with 'operator'.

**Referee:**

L137: This is interesting, please describe how this worked in detail. What if operator 2 found an error? Or a missing grain? How was this dealt with?

**Response:**

We introduced more details in the new MS (L139-L142).

'To ensure the quality of manual labels, a cross-check labeling workflow was used. When an operator finished labeling an image, the labels would be double-checked by the other operator (the inspector), the missing grains found by the inspector would be confirmed by both two operators, and only those consensus 'missing grains' would be added to labels.'

**Referee:**

L138-151: It is great to see some inter-rater agreement work being done here. But I am having trouble interpreting the results. One option is to use a more standard ML metric for segmentation tasks, like intersection over union. Another idea is to explain what the reader should understand from those errors: how much actual error is there between labelers? How much does it change grain size measurements? I think there needs to be a sentence to help a reader get intuition on this metric and these numbers.

**Response:**

We rewrote the paragraph (L151-L159) to help explain the result.

'As shown in Fig. 3, the five operators showed consistent median, first/third quantile and maximum/minimum values for all $D_{normalized}$ statistics and all grain percentiles, indicating the consistent predictive ability of the five operators for grains in diverse environments. An exception is $D_{50}$, in which operators 2 and 5 showed a higher maximum value of $D_{normalized}$ than the other three operators. The inconsistency for $D_{50}$ prediction mainly arises from the predictions for the three photos from the SAFL dataset in which the bed contains a lot of fine grains, and operators 2 and 5 overestimated the $D_{50}$ by merging fine grains as larger sediments. The analysis suggests operator 1 produces consistent grain size for all percentiles, but operator 2 may overestimate $D_{50}$ for images with fine grains. Overall, the manual labels datasets prepared by operators 1 and 2 were consistent with labels created by human operators.'

**Referee:**

L156: Please cite the software library

**Response:**
The software library has been cited (L165).

**Referee:**
L158: what does 'better overall performance' mean, specifically?
**Response:**
We rephrased the sentence: 'the Sigmoid contrast filter was chosen for its lowest predictive error.'.

**Referee:**
L179: I think a paragraph is missing that described how a mask from UNET was converted to actual grain size metrics?
**Response:**
Agreed. The paragraph for grain size calculation was originally in section 3.4 regarding model evaluation (L206-L211 in the old MS), and we moved the paragraph here as a part of the model framework (L187-L192).

**Referee:**
Section 3.2.2: Please discuss more details about the implementation: what ML framework did you use? What were the hyper parameters: Loss function, the optimizer, the learning rate, the batch size, if you chose a weight initializer, how many epochs, the callbacks/ stopping conditions, how many filters were used in the encoding and decoding layers, and how many total tunable parameters were in the model. Also what was the training/validation split, and was it random?
**Response:**
We added a paragraph (L200-L209) to elaborate the U-net implementation.
'The U-net was implemented based on a python library *pytorch* (Paszke et al., 2019). The cross entropy loss function and the stochastic gradient descent were used for model optimization. Model hyperparameters were tuned based on grid searching optimization and 5-fold random cross validation (Goodfellow et al., 2016). The training speed for *U-net* is influenced by the number of images in the training datasets, the batch size and the number of training epoch. Given a fixed training datasets, the hyperparameter *number of training epoch* was tuned first, followed by the *learning rate*. The maximum *batch size* depends on GPU memory and we preferred larger *batch size* for faster training speed when several *batch size* values result in a similar error during the cross-validation. The optimized model hyperparameters are: (1) *number of training epoch* = 150; (2) *learning rate* = 0.005; (3) *batch size* = 96; and (4) *image tile size* = 512. The optimum image tile size was determined based on the analysis in section 5.2.'

**Referee:**
L203-205: please explain the calibration procedure, and each of these parameters.
**Response:**
We revised the paragraph to better introduce the key parameters (L220-L229). We also added a paragraph to elaborate on the calibration of BASEGRAIN (L230-L237).

**Referee:**
Section 4.3: have these images been seen by the UNET? Are they a hold-out set?

**Response:**

The test set is a hold-out set and all the results in section 4 are based on images that have not been seen by the model. The importance of section 4.3 are that (1) the images in section 4.3 are airborne images while the model is trained by terrestrial images, and (2) the images in section 4.3 are from a totally different site with a totally different environment from the training images. The analysis on these un-calibrated images is important to test the robustness and applicability of our model to new environments and to potential watershed-scale drone surveys.

We added more information in L254 – L256 to address the importance of section 4.3.

**Referee:**

L294-303: Please explain these errors. You might give examples.

**Response:**

Has been revised (L328-L335).

**Referee:**

L308-309: Please explain more, add a citation if one exists, and/or present data if you have it.

**Response:**

We don't have systematic data to quantify the relationship between operator fatigue and grain labeling error as this observation was based on the qualitative experience of our operators. We added more text to explain the sentence (L345-L346).

'based on the experience of all five operators, when the operators get tired after hours of labeling work, the labeling error usually increases (especially for fine grains) with operator fatigue.'

**Referee:**

Section 5.1: The use of acronyms and symbols in this section makes it difficult to read. The same is true for other discussion/conclusion sections, with respect to the Error metrics.

**Response:**

We revised the discussion section to make it easier to read. We used the full names in place of acronyms and symbols when they appear for the first time in Section 5.1 and other discussion/conclusion sections.

**Referee:**

L331-333: these lines makes me think that perhaps you could have gotten a better result if you optimized the parameters better, or consulting an expert on BASEGRAIN. Please discuss how you optimized the parameters or how you chose the settings you used to get the best possible results? I think it is important to do this, since ML requires many design decisions and hyper parameter tunings too.

**Response:**

In L224-L237, we added a section to better describe the decisive parameters and the specific parameter tuning procedures for BASEGRAIN calibration.

We also revised the text in section 5.1 to introduce the limitations of BASEGRAIN in greater detail (L365-L368).

**Referee:**

In this part of the manuscript and in general there is a significant focus on why this method was better than basegrain specifically. I know there are many other automated grainsize detection routines, so I wonder if the manuscript can instead position itself to discuss the strengths of the GrainID method generally, instead of how good the technique is compared to BASEGRAIN. This comment is for this Line, but is a general comment for the manuscript as a whole.

**Response:**

Thank you and we have accepted your comments.

To better address the strengths of GrainID, we revised the results section (section 4, L252-L256, L298-L299 and so on). The analysis in the results show that GrainID shows comparably high performance with manual labeling when compared to sieving results (section 4.1) for flume experiments, and much better performance for all grain percentiles for images from a variety of environments (the whole test dataset) than BASEGRAIN and Wolman counts (section 4.2). GrainID also shows the robustness of the program when applied to environments with organic debris and inter-granular noise (section 4.4), and images from a different environment (uncalibrated rivers) and different photography methods (airborne photos) compared to the images in the training dataset (section 4.3). Also, we revised the manuscript to better introduce the splitting of datasets (L119-L122) and the calibration process of GrainID (L201-L209).

Meanwhile, we largely revised Section 5.1 by addressing the advantages of the GrainID model structure, and discussed how GrainID performed in our analysis in greater detail. In Section 5.4, we introduced GrainID as an efficient grain sizing method by addressing the time consumption for GrainID model training and predicting, and the potential of our model for future use considering the fast development of photography and GPU computation techniques.

We realized the importance of elaborating upon the strengths of GrainID. However, as the first study to apply the machine learning image segmentation method U-net on grain detection, our focus is to test the applicability of this method in grain detection by comparing its predictive ability with existing methods for a large dataset. There are some limitations of our study, e.g. the choice of the original GrainID model, and the limited size of training datasets considering the wide range of environmental noise (e.g. lighting, organic debris, water …) when applied to the field. We further discussed theses limitations in the manuscript, and we will try to improve the model in the future.

**Referee:**

Section 5.4: I appreciate that the trained unet takes 5-22 seconds to run for each sample with a GPU. However, I do think the manuscript needs to discuss how long it took to train the model. Keep in mind that BASEGRAIN and Pebble counts do not require model training. Additionally, picture methods require collection and analysis back at a lab, vs being able to count grains in the field.

**Response:**

Agreed.

We realized the importance to discuss the time consumed in training in Section 5.4. We revised Section 5.4 as well as Methods (L203-L207) to better introduce the consumption of time in GrainID for model training. We replaced the term 'model efficiency' with 'predicting time' for a more accurate description. GrainID requires a significant amount time for cross-validation (~ 40 hours for GTX 1080Ti) and model training (~10 hours for GTX 1080Ti), while BASEGRAIN and Wolman count methods do not require model training.

For model efficiency, the advantage of GrainID lies in that (1) due to the robustness of the model, if the machine learning model is trained based on a sufficiently large dataset, the model can be directly used for a new grain size survey without specifically training for the survey region; and (2) for predicting a large dataset (thousands of images), the advantage of GrainID's ability to process numerous images is significant despite the need for model training.

We use image-based Wolman Pebble Count methods in our study.

We think a very important advantage of the machine learning model is the robustness of the model. Considering recent advances in photography and GPU computation techniques, we think that in the future GrainID trained on a sufficiently large dataset can be directly used for many grain size surveys without specifically training for the target study region. We elaborated on the robustness of GrainID in Section 4.3.

**Referee:**
L396: so these images were never seen with the model?
**Response:**
The images used in the results section were never seen with the GrainID model.

**Referee:**
L410-412: Please clarify the purpose of these lines.
**Response:**
We revised the sentence to make it easier to read.

**Referee:**
Section 5 generally, and L413-417 specifically: Can this section be generalized for any photographic analysis technique for grain size? Can the manuscript present any general things that others may use?
**Response:**
We revised section 5 and L459-L464 to discuss the potential future development of general grain sizing methods with advances in photography and GPU computation techniques in the future.

**Referee:**
Figure 3: Can the manuscript explain a bit more what practical insight the reader should get from this figure? I don;t have much intuition for the metric.
**Response:**
We rewrote the paragraph (L151-L159) to help explain the result.

'As shown in Fig. 3, the five operators showed consistent median, first/third quantile and maximum/minimum values for all $D_{normalized}$ statistics and all grain percentiles, indicating the consistent predictive ability of the five operators for grains in diverse environments. An exception is $D_{50}$, in which operators 2 and 5 showed a higher maximum value of $D_{normalized}$ than the other three operators. The inconsistency for $D_{50}$ prediction mainly arises from the predictions for the three photos from the SAFL dataset in which the bed contains a lot of fine grains, and operators 2 and 5 overestimated the $D_{50}$ by merging fine grains as larger sediments. The analysis suggests operator 1 produces consistent grain size for all percentiles, but operator 2 may overestimate $D_{50}$ for images

with fine grains. Overall, the manual labels datasets prepared by operators 1 and 2 were consistent with labels created by human operators.'

**Referee:**
Figure 4d: I do not think the UNET implementation presented here has any 'Crop' operations, just 'copy' (green arrow)? I think the red arrows should be ConvTranspose (or upsample), and not pool?
**Response:**
Agreed, it was a typo which has been corrected.

**Referee:**
Table 1: Training column does not sum to 136 images for me. Also what about validation?
**Response:**
To train the machine learning model to better distinguish sediments from field environmental elements and improve the model robustness, we specifically collected 42 field photos primarily consisting of various environmental elements (e.g. cohesive sands, wood, vegetation and water) with limited sediment grains in the images in addition to the 94 images in the training column of Table 1. In the revised manuscript, we added these 42 images to the training column of table 1, and we added a section to explain why we used these 42 photos to train the model (L95-L98).
We realized we should introduce the cross-validation more clearly, so we added more details for the split of the dataset ('train+validation/test') at L119-L122, and for the cross-validation methods (L200-L203).

---

## Author Response (AR2)

**Response file to comments**

**1. Response to referee #1, Byungho Kang**

**Byungho Kang:**
The test used in this study for validating the method is indeed impressive.

It would be better to include some information about the size of original images used for training and validation (though it seemed to have varying image sizes depending on the dataset) and add a little about how the method could work adaptively on different image sizes.

**Response:**
Thanks. **We added the original image size information in L95 – L96.**
Yes, the image sizes vary depending on the dataset. The end-to-end image-based grain size measurement depends on not only the grain detection algorithms (like our GrainID), but also the collection of grain images. The important questions include (1) what image resolution/size we should use for a specific flume/field measurement, and (2) how the image resolution/size influence the predictive ability of our GrainID model.
For the first question, the image resolution depends on your research need, the finer grain to be detected, the higher resolution needed. The image size should be large enough to capture the grain size distribution even with the presence of environmental elements in the image (L137).
For the second question, to account for the varying input image size, we split the original images into image tiles, and applied the overlap-tile strategy to compensate the split error and achieve seamless prediction (L170-L179). In section 5.2, we discussed the influence of image tile size on grain detection ability by varying the image tile size (64*64, 128*128, 256*256, 512*512, 768*768, 1024*1024) while maintaining the raw image resolution. In addition, we discussed the influence of image resolution on grain size detection by down-sampling the original image resolution of 0.45 mm/pixel up to 4.5 mm/pixel and comparing the results of down-sampled images to the sieving results.
The analysis above shows how the image resolution and tile size influence the predictive ability of GrainID. However, we realized that the analysis in Section 5.2 is based on a small dataset, which limits the application to greater variety of environments. **We added content in section 5.5 (L456 - 457) to address this limitation in the revised manuscript.**